# Environmental Citizenship and Energy Efficiency in Four European Countries (Italy, The Netherlands, Switzerland and Hungary)

**Ibolya Czibere [1], Imre Kovách [1,2] and Gergely Boldizsár Megyesi [2],***

[1] Department of Sociology and Social Policy, University of Debrecen, Egyetem tér 1, 4032 Debrecen, Hungary; czibere.ibolya@arts.unideb.hu (I.C.); kovach.imre@tk.mta.hu (I.K.)

[2] Institute for Sociology, Centre for Social Sciences, Hungarian Academy of Sciences, Tóth Kálmán u. 4, 1095 Budapest, Hungary

* Correspondence: megyesi.boldizsar@tk.mta.hu

**Abstract:** In our paper we aim at analysing the social factors influencing energy use and energy efficiency in four different European countries, using the data from the PENNY research (Psychological social and financial barriers to energy efficiency—Horizon 2020). As a part of the project, a survey was conducted in four European countries (Italy, The Netherlands, Switzerland and Hungary) to compare environmental self-identity, values and attitudes toward the energy use of European citizens. Previous research has examined the effect of a number of factors that influence individuals' energy efficiency, and attitudes to energy use. The novelty of our paper that presents four attitudes regarding energy use and environmental consciousness and compares them across four different regions of Europe. It analyses the differences between the four attitudes among the examined countries and tries to understand the factors explaining the differences using linear regression models of the most important socio-demographic variables. Finally, we present a typology of energy use attitudes: four groups, the members of which are basically characterised by essentially different attitudes regarding energy use. A better understanding of the diversity of energy use may assist in making more accurate policy decisions.

**Keywords:** cross-country comparison; environmental attitudes; values and efficiency; sustainability; socio-demographic characteristics

## 1. Introduction

This study is a cross-country assessment of the links between household energy use (residential energy use), environmental self-identity, the attitudes toward energy use and socio-demographic variables associated with energy use. Household energy use and the related energy savings of the industrial and service sectors through using renewable sources may be essential elements of sustainability. Direct household energy demand accounts for a quarter of the EU's energy use [1], which is even a higher proportion, close to 40% if we include the energy consumption of the industry and services serving household consumption. This sector, in particular, offers a lot of chances for energy efficiency and cost-efficient energy use and thus for the increase of sustainability.

The energy consumption of households is connected with sustainability in various areas of consumption. In the literature, consumer behaviour affecting direct energy use, such as heating, cooking, household appliances, transportation, automobile use [2–5] or the renovation and insulation of housing [6–8], food consumption and the implications of production for self-consumption [9] are all linked to the issues of sustainability, environmental consciousness, energy use and climate change [10].

Sustainability opportunities are increased by activities promoting or inhibiting residential energy efficiency [11–15], but the research covers various social areas (domains) of this complex problem. The central research question of some of them concerns the behaviour of household consumers. The role of access to information and knowledge in shaping consumers' energy consumption [16,17], government actions, local models [18,19], the motivating effect of good examples [20], the impact of incentives and key players in the dissemination of sustainable patterns of behaviour [21,22], in addition to the role of forms of knowledge [23] the effect of reference groups, acquaintances and friends' behaviour [3]. Another round of research relates to the correlations between disparities and energy efficiency/sustainability, such as investment opportunities [4], quality of life, well-being and energy efficiency [24], the topic of energy poverty [25–27], and correlations between economic development and energy efficiency [28,29]. Bhattacharjee and Reichard's study provides an overview [30] of how socio-economic factors reduced the energy use of residential customers. Gram and Hanssen [31] analyse the role of consumer habits, amongst the factors contributing to lower energy consumption.

Our study aims to use the database of an international research project and contribute to exploring the barriers of energy efficiency and using renewable energy sources, strengthening the chance of a sustainability transition and to discover the differences between regions using the example of four European countries, a Mediterranean one: Italy, an Alp one: Switzerland, a Western European one: the Netherlands and a Central-Eastern-European one: Hungary, thus providing opportunities for comparing regions with different social backgrounds. During this analysis, four attitudes regarding energy use and the environment protection were separated and, based on these attitudes, we have created four groups (clusters) of energy use attitudes.

## 2. Barriers to Energy Efficiency: Literature Background

Cristina Cattaneo presented a summary analysis on the barriers to energy efficiency and the possible role of political interference [2], which served as the foundation of a research project that enabled the creation and analysis of the database used in this study. Based on the literature, she describes two domains of consumer behaviour in the context of energy efficiency. The first domain is the behaviour related to the repeated reduction of energy use as a daily routine. The second domain is the behaviour related to long-term energy efficiency investments. Cattaneo stresses that studying barriers, despite all the research results, continues to be justified.

Variants of energy-efficient behaviours differ in line with the temporal, risk and environmental preferences of individuals according to the paper by Schleich, Gasmann, Faure et al. [32], which is the one most accepted by the literature on organising barriers. Schleich et al describe internal and external barriers. Consumers underinvest in energy-efficient technologies [33], and Schleich and others include those factors in the theoretical framework of the internal and external barriers that could explain the low adoption behaviour of agents in the context of energy efficiency.

External barriers include factors that restrict the introduction of energy-efficient technologies. These are easily changeable factors, primarily depending on the institutional environment, which are usually summarised as market-failure explanations [34]. The external barriers, in Schleich et al.'s system, are capital market system failures, lack of information, asymmetric, ambiguous information and financial and technological risks. The lack of information is the greatest barrier to external (political) interference [35], but the low level of presentation in the market of products promoting energy efficiency or if the buyer is unable to ascertain the effectiveness of lowering the energy usage of the product are also significant. Cattaneo stresses that according to the observations of behavioural economics and psychology, the behaviour of agents rarely meets the criteria for rational decision-making and it is difficult to distinguish the consequences of one behavioural factor from those of the others [36,37].

According to the model of Schleich and others [32], external barriers are independent of the actors making decisions on energy use, and these barriers may result from institutional circumstances and failures of the market. Capital market imperfections or fragmentary information about the product and the availability of energy-efficient technologies are also external barriers. The external barriers,

particularly the lack of relevant information, could lead to keeping energy-efficiency investments at low levels [38,39]. In our paper, we deal with differences in internal barriers to household energy use, according to countries. Analysis of the differences between the countries, although it may give an evident explanation for understanding internal barriers, receives relatively little attention.

## 3. Materials and Methods

The aim of the PENNY project was to examine social, psychological, economic and financial factors affecting energy efficiency, using a large-scale survey study. The first fieldwork was undertaken in Switzerland, Italy and the Netherlands, followed by Germany and Hungary in 2017 and 2018, respectively as Tables 1 and 2 shows. The online interviews were conducted in cooperation with service providers in three countries (in Switzerland: Aziende Industriali Lugano and Stadtwerk Winterthur, in the Netherlands: Qurrent, in Italy: ENI,). Five-thousand three-hundred and eight households in four countries participated in the survey study. Of the respondents, 28.4% took part through a telephone interview, while 71.6% through online completion.

**Table 1.** Breakdown of respondents by country.

|  | **Frequency** | **Per cent** |
| --- | --- | --- |
| Switzerland | 1178 | 22 |
| *The Netherlands* | *2213* | *41.3* |
| Italy | 1508 | 28.2 |
| *Hungary* | *453* | *8.5* |
| Total | 5352 | 100 |

Own compilation.

**Table 2.** Interview method.

|  | **Frequency** | **Per cent** |
| --- | --- | --- |
| Phone | 1508 | 28.4 |
| Web-portal | 3844 | 71.6 |
| Total | 5352 | 100.0 |

Own compilation.

The questionnaire was prepared on the basis of the questionnaire developed in the Centre for Energy Policy and Economics at ETH Zurich in the project titled "Underlying energy efficiency and technological change in the Swiss household sector" for the EU H20 PENNY project survey.

Table 3 shows the implementation of large sample surveys. In Switzerland, the survey was conducted in two different cities in the country. The customers of two providers gave the base population: 15,000 customers in Winterthur city and 13,100 customers in Lugano city and its surrounding municipalities, from which a random sample was selected. In the Netherlands, the survey was conducted in Grönigen, a regional centre and university town. The base population consisted of consumers with a smart meter for at least six months. In Italy, the questionnaire was conducted in its highly developed, urbanized and industrial northern region. The sample consists of ENI consumers with a contract allowing the transfer of their data to a third party for research purposes, from which a stratified sample was prepared, based on contract type, consumption history and place of residence. In Hungary, a random sample of Internet subscribers from Debrecen and Hajdú-Bihar County answered the questions. This is a semi-urbanized region with a university town. The representativity of the samples could not be ensured in advance and therefore that of all respondents can be interpreted in comparison with the national data for some of the characteristic features below.

**Table 3.** Implementation of the large sample survey [1].

| Country | Switzerland | The Netherlands | Italy | Hungary |
|---|---|---|---|---|
| No. of homes contacted | 28,100 | 19,000 | 102,000 | 121,133 |
| Means of contact | postal letter | e-mail | e-mail | Facebook |
| Population: | Customers of 2 public utility providers (Lugano and Winterhur): | Customers, having a contract for more than 6 months with smart | ENI customers providing explicit and written consent to take part in the research. | People living in Debrecen and Hajdú-Bihar County |
| Sample | Random | Random | Stratified sample of ENI consumers, representative for location, consumption patterns and contract characteristics | Random |

Source: Boogen and Daminato [40] and our own compilation. [1] In our study, the German data were not used due to the low sample size.

The average age of the respondent is 51 years. The Dutch population is slightly younger (48.9 years), the Swiss and Italian population is slightly older: 52 and 53 years old, while the Hungarian average age is 51.8 years.

Fifty-six per cent of respondent households in Italy and 48 per cent in Switzerland live in a multi-family house. Almost three-quarters of Dutch households and 70% of Hungarian respondents live in a detached property. The proportion of homeowners in the sample, with the exception of Hungary, is overrepresented compared to national statistics (between 59 and 85%). Ninety per cent of Hungarian households live in their own homes, and the proportion of Hungarian homeowners in the sample is close to that, 86%. In the EU-27, 71% of inhabitants live in their own property, as shown in Table 4.

**Table 4.** The nature of respondents' homes.

| Respondent's Country | Single-Family Detached House | Semi-Detached House | Terraced House | Apartment in a Multi-Family House | Total |
|---|---|---|---|---|---|
| Switzerland | 316 | 170 | 121 | 569 | 1176 |
| % | 26.90% | 14.50% | 10.30% | 48.40% | 100.00% |
| The Netherlands | 377 | 363 | 875 | 591 | 2206 |
| % | 17.10% | 16.50% | 39.70% | 26.80% | 100.00% |
| Italy | 356 | 174 | 128 | 850 | 1508 |
| % | 23.60% | 11.50% | 8.50% | 56.40% | 100.00% |
| Hungary | 267 | 10 | 13 | 119 | 409 |
| % | 65.30% | 2.40% | 3.20% | 29.10% | 100.00% |
| Total | 1316 | 717 | 1137 | 2129 | 5299 |
| % | 24.80% | 13.50% | 21.50% | 40.20% | 100.00% |

Source: own compilation.

The average monthly gross income of households in Italy and the Netherlands is between EUR 1500 and EUR 4000, in Switzerland between EUR 5500 and EUR 8000, while in Hungary, it is below EUR 1500. This is consistent with the countries' median household income data, according to OECD statistics.

In Table 5, the qualification of the respondents is presented. According to the data, it is significantly higher than that shown by the national statistics for each country [40]. In particular, Hungarian data is far above the national average. Tertiary education ranges from 35% in Italy to around 70% in the Netherlands.

**Table 5.** Qualification of the respondents.

| | Less Than Elementary School | Lower Secondary School or Vocational School | Upper Secondary School | MA or BA | Post-Graduate | Total |
|---|---|---|---|---|---|---|
| Switzerland | 7 | 345 | 110 | 550 | 74 | 1086 |
| | 0.60% | 31.80% | 10.10% | 50.60% | 6.80% | 100.00% |
| The Netherlands | 33 | 396 | 145 | 1233 | 107 | 1914 |
| | 1.70% | 20.70% | 7.60% | 64.40% | 5.60% | 100.00% |
| Italy | 14 | 271 | 702 | 465 | 56 | 1508 |
| | 0.90% | 18.00% | 46.60% | 30.80% | 3.70% | 100.00% |
| Hungary | 7 | 45 | 115 | 112 | 5 | 284 |
| | 2.50% | 15.80% | 40.50% | 39.40% | 1.80% | 100.00% |
| Total | 61 | 1057 | 1072 | 2360 | 242 | 4792 |
| | 1.30% | 22.10% | 22.40% | 49.20% | 5.10% | 100.00% |

Source: own compilation.

According to Table 6 almost 27% of the respondents are economically inactive, with the lowest proportion being 23.1% among Dutch respondents and the highest among Hungarians (51.5%). The share of full-time employees is the highest among the labour market groups, at 45.1%. Of the sample, 92% are decision-makers in the household. Men account for 63.4% and women for 36.4%.

**Table 6.** Labour market status of respondents.

| | Full Time | Part-Time | Entrepreneur | Retired | Unemployed | Student | Other | Total |
|---|---|---|---|---|---|---|---|---|
| Switzerland | 411 | 232 | 104 | 249 | 12 | 23 | 32 | 1063 |
| | 38.70% | 21.80% | 9.80% | 23.40% | 1.10% | 2.20% | 3.00% | 100% |
| The Netherlands | 832 | 346 | 190 | 323 | 61 | 41 | 40 | 1833 |
| | 45.40% | 18.90% | 10.40% | 17.60% | 3.30% | 2.20% | 2.20% | 100% |
| Italy | 701 | 65 | 152 | 370 | 42 | 9 | 51 | 1390 |
| | 50.40% | 4.70% | 10.90% | 26.60% | 3.00% | 0.60% | 3.70% | 100% |
| Hungary | 109 | 8 | 9 | 100 | 13 | 25 | 4 | 268 |
| | 40.70% | 3.00% | 3.40% | 37.30% | 4.90% | 9.30% | 1.50% | 100% |
| Total | 2053 | 651 | 455 | 1042 | 128 | 98 | 127 | 4554 |
| | 45.10% | 14.30% | 10.00% | 22.90% | 2.80% | 2.20% | 2.80% | 100% |

Source: own compilation.

The questionnaire has three main modules. The first module collects the characteristics of the dwellings and the social, demographic and economic characteristics of households. The second module includes measurements of biospheric, altruistic, egoistic and hedonic values, as well as environmental self-identity, personal norms, corporate environmental responsibility and social norms, according to Steg, Keizer, Ruepert [41] and Schwartz and Howard [42]. The purpose of our study is to analyse the latter. The third module was edited to measure the level of knowledge about electricity costs, financial literacy, and the lifetime costs of two alternative appliances.

## 4. The Aim of the Study

The aim of our research is to analyse the social and demographic factors of environmental self-identity, personal norms, corporate environmental responsibility and social norms, interpreted as barriers to energy efficiency and renewable energy use. Our research question is the extent to which the responses given are interrelated, the social and demographic characteristics that influence the responses and their connections, and to what extent these help or weaken the use of renewable energy use and energy efficiency. This study focuses on consumers because their lack of interest may disrupt vertical market relationships and this may be an insurmountable obstacle to increased use of renewable energy.

The variables in Table 7 were used to create each principal component. Respondents were able to agree on a scale of 1 to 7 with each statement with 7 being the strongest agreement. The following tables present the averages of the variable assemblies that make up the principal components by country.

**Table 7.** Mean and standard deviation of energy use variables by country [2].

| Country | Ch | | NL | | I | | Hu | | Sum | |
|---|---|---|---|---|---|---|---|---|---|---|
| | Average | Deviation | Average | Deviation | Average | Deviation | Average | Deviation | Average | Deviation |
| **Environmental Self-identity Barrier** | | | | | | | | | | |
| Acting pro-environmentally is an important part of who I am. | 5.30 | 1.36 | 4.99 | 1.45 | 6.21 | 1.09 | 6.07 | 1.20 | 5.50 | 1.41 |
| I am the type of person who acts pro-environmentally. | 5.32 | 1.14 | 5.15 | 1.22 | 5.97 | 1.13 | 5.69 | 1.22 | 5.47 | 1.23 |
| I see myself as a pro-environmentally person. | 5.36 | 1.19 | 5.13 | 1.22 | 6.12 | 1.08 | 5.57 | 1.42 | 5.51 | 1.26 |
| **Energy-provider Environmental Responsibility Barrier** | | | | | | | | | | |
| I think my energy provider has a goal to minimise its impact on the environment. | 5.15 | 1.50 | 5.37 | 1.44 | 5.80 | 1.36 | 4.04 | 1.83 | 5.36 | 1.52 |
| I think my energy provider has implemented policies and procedures to minimise its impact on the environment. | 5.04 | 1.54 | 5.21 | 1.47 | 5.51 | 1.47 | 3.86 | 1.75 | 5.17 | 1.56 |
| I think my energy provider has stated its mission to implement a sustainable (pro-environmental) policy | 5.55 | 1.25 | 5.48 | 1.53 | 5.69 | 1.42 | 3.94 | 1.92 | 5.46 | 1.52 |
| **Personal Norms Barrier** | | | | | | | | | | |
| I feel morally determined to save energy | 4.33 | 1.39 | 5.87 | 1.12 | 4.94 | 1.58 | 5.49 | 1.56 | 5.22 | 1.50 |
| It is my moral ideal to save energy | 4.67 | 1.61 | 5.42 | 1.33 | 4.82 | 1.55 | 5.98 | 1.49 | 5.10 | 1.53 |
| I would act according to my principles if I save energy | 3.47 | 1.73 | 5.47 | 1.37 | 5.02 | 1.64 | 5.84 | 1.50 | 4.91 | 1.75 |
| I feel personally responsible for trying to save energy | 5.52 | 1.28 | 5.61 | 1.27 | 5.67 | 1.43 | 5.85 | 1.42 | 5.62 | 1.33 |
| **Social Norms Barrier** | | | | | | | | | | |
| Most of the people who are important to me think I should try to use as little energy as possible | 4.84 | 1.40 | 3.79 | 1.66 | 4.29 | 1.88 | 4.92 | 1.86 | 4.25 | 1.75 |
| Most of the people who are important to me will approve of when I try to use as little energy as possible | 4.71 | 1.37 | 5.55 | 1.34 | 5.30 | 1.51 | 5.35 | 1.70 | 5.27 | 1.46 |
| Most people who are important to me try to use as little energy as possible | 5.00 | 1.35 | 4.41 | 1.33 | 4.87 | 1.56 | 5.14 | 1.73 | 4.73 | 1.46 |

Source: own compilation. [2] Don't know answers were treated as missing answers.

Table 8 presents the Cronbach Alpha's of the four sets of variables, and according to it, each set is fit to prepare a principal components analysis.

In our paper, we tried to capture four attitudes related to energy use, and accordingly, we created four principal components and indices. The first major component measures environmental identity, the second the role of personal norms in energy use, the third the role attributed to energy providers, and the fourth the role of social norms in shaping attitudes influencing energy use. Individuals with

strong environmental self-identity see active participation in environmental activities as a determinant of their identity [43]. Personal standards refer to how people feel about their moral commitment to energy-saving behaviour [42,44]. Corporate environmental responsibility means increasing the environmental performance of organisations and reducing their environmental impact [41]. Social norms, on the one hand, including how people considered as reference reduce their energy use and how they think about what an individual can do [45].

**Table 8.** Average and standard deviation of country-specific indices of energy use attitudes.

|  | IT | NL | CH | HU |
|---|---|---|---|---|
| **Mean (Standard Deviation)** | | | | |
| Environmental self-identity | 6.10 (1.03) | 5.09 (1.18) | 5.33 (1.10) | 5.77 (1.16) |
| Personal norms | 5.67 (1.28) | 5.42 (1.19) | 5.33 (1.15) | 5.78 (1.28) |
| Energy-provider environmental responsibility | 4.93 (1.49) | 5.59 (1.14) | 4.14 (1.25) | 3.94 (1.70) |
| Social norms | 4.83 (1.39) | 4.58 (1.16) | 4.87 (1.23) | 5.13 (1.57) |
| **Cronbach's Alpha** | | | | |
| Environmental self-identity | 0.93 | 0.89 | 0.87 | 0.894 |
| Personal norms | 0.92 | 0.86 | 0.85 | 0.881 |
| Energy-provider environmental responsibility | 0.93 | 0.87 | 0.7 | 0.925 |
| Social norms | 0.78 | 0.71 | 0.88 | 0.871 |

Source: Boogen and Daminato 2019 [40] and own compilation.

## 5. Results

### 5.1. Environmental and Energy Use Attitudes

In the following, we will analyse the primary differences between the four principal components related to energy use and environmental awareness in each country, and their differences across socio-demographic variables.

#### 5.1.1. Differences by Country

When evaluating *individuals' environmental responsibility*, we examined whether environmental awareness is important for individuals, whether they live in an environmentally friendly way, and consider themselves to be environmentalists. In this field, the results are the most positive in Italy and Hungary, and they are much more committed to environmental responsibility than the Dutch or Swiss respondents. The results in Italy show a particularly strong self-image. The Swiss results express criticality, but the results are close to the average. The most divergent to negatives tend to be the Dutch opinions, results express a clear negative, rejecting attitude, they consider themselves to be the least environmentally conscious, for whom environment consciousness is not important and they do not live in an environmentally conscious way, as Table 9 shows.

In the domain of attitudes towards *the environmental responsibility of energy companies*, on the other hand, the Swiss and the Dutch are generally satisfied with their attitudes towards the environment, but the Swiss tend to have a negative view of the activities of these organisations in terms of protecting the environment and minimising their impact on the environment. Italian respondents are the most satisfied with the environmental activities of energy suppliers. Based on the Hungarian responses, satisfaction with Hungarian organisations lags far behind the functioning of organisations in other countries, and in this sense, there is strong distrust in Hungary towards service provider organisations.

With the *environmental responsibility moral dimension*, we looked at individuals' attitudes, in particular, how much of a moral, principled, and personal responsibility issue it was for them to use energy sustainably. The Swiss respondents are the most critical in this area, and they describe their own

sense of responsibility in the most negative way. In Italy, individuals rate themselves as average but tend to consider their own attitudes as negative. The Dutch value it highly, and Hungarian respondents also say they are extremely committed.

**Table 9.** Differences of principal components by country.

| Countries | Environmental Self-Identity | Energy-Provider Environmental Responsibility | Personal Norms | Social Norms |
|---|---|---|---|---|
| Switzerland | −0.1333525 | −0.0632440 | −0.5776945 | −0.0611805 |
| Netherlands | −0.3277172 | 0.0161818 | 0.3311583 | −0.1272072 |
| Italy | 0.5065005 | 0.2529796 | −0.0912623 | 0.0539995 |
| Hungary | 0.2297430 | −1.0186736 | 0.4781927 | 0.2922530 |

Notes: (sig = 0.000); Source: own compilation.

*With the social dimension of environmental responsibility*, we measured how important it is for individuals to use energy sustainably. Respondents in Switzerland, Italy and Hungary feel that sustainability as important to the people with whom they are related or who are important or important to them, and it is characterised by average, but positive values. The Dutch also think similarly, they value their own personal relationships along with the average but tend to be slightly negative. Hungarian results are the best. Hungarian respondents characterise their environment with slightly above average environmental awareness and sustainable energy use.

If *personal (moral) dimensions are compared with environmental (social) dimensions*, we find that the Swiss consider the behaviour of their social environment much more positively than their own, and they seem to perceive their reference group as being more responsible and aware compared to themselves. The Italian respondents are surrounded by a typical social environment that behaves similarly to themselves. The Dutch consider themselves very positive and above average, while their environment is perceived as average, and even slightly more negative, while the Hungarian respondents consider themselves and those around them equally above average and highly committed in the field of environmental sustainability.

In the field of *differences between countries*, in Switzerland, respondents were dissatisfied with almost all dimensions, with average or above-average negative ratings, highly critical of themselves, with the exception of people in their environment who were perceived to have a moderately positive attitude towards themselves. In the Netherlands, they are very critical of their own environmental responsibility, they also tend to have a negative view of their environment but less critically, although the results are rather close to average.

In Italy, most dimensions were rated positively, especially individual and organisational behaviours. In both cases, they tend to consider the attitudes of themselves and the people surrounding them to be average, though they tend to view their own assessment negatively. In Hungary, all values are more positive than the average except for their opinion of energy suppliers, to which Hungarian respondents reacted very critically.

5.1.2. Differences according to Household Size

According to the results, the size of the households is irrelevant in most cases to the principal components, except for the "Own Environmental Responsibility" dimension as shown by Table 10.

In the whole sample, the environmentally-conscious behaviour of individuals may be considered as average, slightly differently in households with 1 and 3 persons, and slightly positive in households with 2 or 4 persons. This is likely to indicate that singles and households with children, on average, pay attention to environmental protection and eco-friendly behaviour and are slightly negative or slightly positive, but nowhere is this marked by over-dissatisfaction or over-satisfaction.

<p style="text-align: center">**Table 10.** Household size—My own environmental responsibility.</p>

| How Many People Lived in your Household in 2017? (Persons) | Environmental Self-identity |
| :---: | :---: |
| 1 | −0.0146606 |
| 2 | 0.0091501 |
| 3 | −0.0020438 |
| 4 or more | 0.0056455 |

<p style="text-align: center">Notes: (sig = 0.000); Source: own compilation.</p>

### 5.1.3. Differences according to Educational Qualifications

Qualifications affect all major components except the "Environmental Responsibility of Energy Companies" key component, as Table 11 shows.

<p style="text-align: center">**Table 11.** Differences according to qualifications amongst the 3 principal components.</p>

| Highest Level of Education Completed | Environmental Self-Identity | Personal Norms | Social Norms |
| :---: | :---: | :---: | :---: |
| Basic education or less | 0.1168013 | 0.3134117 | 0.2715644 |
| High school without graduation | 0.0332096 | 0.0083213 | 0.0911763 |
| High school graduation | 0.2700603 | −0.0272575 | 0.0722657 |
| Degree | −0.1279975 | 0.0032698 | −0.0845552 |
| Postgraduate | −0.0569258 | −0.1154312 | 0.0254983 |

<p style="text-align: center">Notes: (sig ≥ 0.041); Source: own compilation.</p>

For each of the three principal components, average values were obtained for all qualifications. In particular, we find differences in slightly negative or slightly positive ratings.

Those with a basic education value all dimensions positively and even the moral attitude of themselves and their environment is slightly above the average. On the other hand, they are in line with the average terms of environmentally friendly behaviours, which also means their own active involvement. Those with the lowest levels of education had the highest average scores.

The opinion of those without a baccalaureate i.e. skilled workers and vocational school graduates is the most balanced in the positive, all values are in the positive range and also are in line with the average rating.

Graduates appear to be slightly more critical of themselves and are slightly below the average with regard to their personal environmental behaviour and moral principles. At the same time, they view their social environment more positively, associating a more positive image with those who live in their environment than to themselves. On the other hand, they evaluate their personal environmental responsibility and active environmental friendliness above the average.

Graduates are critical in a different way from high school graduates, their opinions have deviated to the negative in two cases. Among them, their own environmental responsibility and the environmental behaviour of the people who are important to them received a more negative evaluation, close to the average. They were less critical of their own moral values, and they characterized it as average but positive. The higher educated seem to be slightly more critical of themselves than the lower educated, with the highest average score given to themselves by the elementary level educated.

Those with postgraduate qualification show some critical attitudes, both in their own environmental responsibility and in their moral judgment. Their scores deviate from the average to the negative, however, compared to themselves, they characterised the behaviour of people living in their environment as average, but in the more positive range.

However, from the direction of the differences according to major components, the results show that the environmentally-conscious self-image, the environmentally friendly lifestyle, and the environmental

behaviour at the upper-end point of qualifications cause some dissatisfaction and a negative deviation from the average. On the moral level, the higher educated tend to be slightly more dissatisfied or more critical, the averages of graduates and postgraduates got close to zero but in the negative range, while the lowest educated feel that they think more responsibly about environmental responsibility than those with higher qualification. The same is true with regard to the individual's social environment, here also those with lower levels of education being the least critical of the environmentally-conscious behaviour of the people who come into contact with them, while they assess it at all other levels as average.

### 5.1.4. Differences according to Employment Status

Table 12 shows the differences in attitudes among employment status. Full-time employees are slightly negative, but on average satisfied with their own environmentally conscious behaviour, they also have a moderate level of trust in energy companies and believe that they provide for counteracting environmental impacts in an acceptable way. They judge their own moral responsibility also in this way and see people in their environment who are important for them as themselves. In this sense, they live in a homogeneous value environment. Respondents who work full-time consider their overall attitude to environmental protection to be nearly average, although all values are in the negative range, indicating that they are not fully satisfied, some criticism and self-criticism are perceivable in their responses.

**Table 12.** Differences according to occupational status amongst the four principal components.

| Employment Status | Environmental Self-Identity | Energy-Provider Environmental Responsibility | Personal Norms | Social Norms |
|---|---|---|---|---|
| Employed (full time) | −0.1019162 | −0.0450327 | −0.0384920 | −0.0583703 |
| Employed (part time) | −0.1499497 | 0.0513203 | −0.0148499 | −0.0536751 |
| Freelancer, entrepreneur | 0.0142916 | 0.0286138 | −0.0136805 | −0.1125472 |
| Pensioner | 0.2574554 | 0.1195596 | 0.0388391 | 0.1821093 |
| Unemployed | 0.1292061 | 0.0495000 | 0.2380619 | 0.1584814 |
| Student | −0.2334114 | −0.4207731 | 0.0216244 | −0.0134410 |
| Other | 0.2112353 | 0.1421837 | −0.0588843 | 0.0860639 |

Notes: (sig. ≥ 0.049); Source: own compilation.

Part-time employees judge most areas relatively negatively, with their responses being in the negative range, with one exception. There is some criticism of their own environmental roles and values, which they rated as average but not positive. They considered only the organisations' environmentally conscious behaviour to be good on average, and in everything else, they view themselves and their environment more critically.

Freelancers and entrepreneurs are not as critical as the previous group, but they not only judged their own environmentally friendly behaviours a bit more critical but average but also characterized those around them with somewhat negative, average behaviours.

Pensioners see their own environmentally conscious behaviour higher, then other groups: they gave themselves the highest average score within each employment group, but this is still closer to the average, though in the positive value range. Responsible behaviour and thinking of those around them were similarly valued. In one single principal component, in their own moral attitudes, we find some critical expression, their values are almost average, but still more positive here.

Of the five main employment groups, students' results differ most significantly from any evaluation so far. Students are the absolute critical group. Beyond their own moral values, they are completely dissatisfied with above-average or near-average negative results in the case of every principal component. In particular, they are very dissatisfied with the environmentally-conscious behaviour of their own and the energy service provider companies, and less so, less critically, on average,

they assess positively what is represented in their own moral values and, in the same way, average, but negatively about their experience in their own environment.

Evaluating the results from the direction of the four principal components, full-time, part-time workers and students were more critical in issues related to their own environmental responsibility, and full-time workers and students with the attitude of service provider organisations In the field of moral expectations towards themselves, the employees (full, part, entrepreneurs) gave a near negative assessment close to the average, while the non-employed (pensioners, students) gave a close to average positive. In the field of moral considerations, also similar results were obtained for the employed and the non-employed. Their personal environment was positively evaluated only by pensioners and the unemployed, with all other employment groups showing more or less dissatisfaction.

The analysis of our results in the domain of environmental consciousness shows a significant difference in the four countries (The Netherlands, Switzerland, Italy, Hungary) in terms of environmental responsibility among their residents. The countries are divided into two distinct subgroups: The Netherlands-Switzerland and Italy-Hungary. The Dutch and Swiss respondents are self-critical and unsatisfied with the level of their environmental consciousness, whereas Italian and Hungarian respondents feel that they are environmentally conscious and have embraced environmentally-friendly lifestyles. Dutch and Hungarian respondents strongly believe that they are committed to protecting the environment, in contrast with the Swiss and Italians, who are not satisfied with their commitment level. The social environment can influence individual behaviour and serve as a reference for values and norms. Studying such factors have resulted in a finding that Hungarian and Italian respondents tend to live in a social environment that is about as environmentally conscious as they are, while the Dutch are not satisfied with those who surround them and consider themselves a more positive example, and the Swiss are generally unsatisfied with their own commitment and view their social environment as a more positive example. Beyond sociodemographic differences, results are also dependent on how informed residents of each country are on the given subject, and how much knowledge they have obtained to make informed individual decisions. Local opinions regarding sustainable behaviours, and how much need there is for sustainability, are influenced by various strategies in each country, targeting specific populations and shaping their views.

*5.2. Factors Influencing Attitudes towards Environmental Protection and Energy Efficiency*

In the following, we present eight linear regressions to test the effects of the different factors of environmental and energy use attitudes.

In Model A1, B1, C1 and D1 we analysed the effect of individual characteristics and characteristics of the home on energy use and environmental attitudes. In the case of each attitude, we built a similar model. We measured personal characteristics by age, gender, employment status, educational level. These latter two variables were dummy variables. We measured the characteristics of the building by the type of the home, whether it is a flat or a house, by the size of the home in square meters and by the number of electronic appliances used in the home.

Table 13 presents the first four model. In Model A2, B2, C2 and D2 we added to the previous model also country variable, as dummy variables to the analysis. The results are presented in the following tables.

The explanation power of Model A1, on analysing the factors influencing environmental self-identity is 8.3%. Among the individual characteristics gender, and age have a significant effect, while all the home characteristics, the type of the household, the size of the home and the number of electronic appliances significantly affects environmental self-identity. According to our data, males tend to keep themselves more environmentally conscious, and by age environmental consciousness is growing. Looking at home characteristics we see that people living in smaller flats with less electronic appliances tend to keep themselves less environmental-friendly persons.

The explanation power of Model B1, on analysing the factors influencing energy-providers' environmental responsibility is 2.5%. Among the individual characteristics gender, and age have a significant effect, while from the home characteristics, the type of the household, and the number of

electronic appliances affect significantly this attitude. According to the analysis, males assume higher the environmental consciousness of the energy providers, than females. This attitude increases by age. Home size and the number of electric appliances significantly affects this attitude. People living in smaller flats and having less electronic appliances tend to see energy providers less environmentally-friendly.

**Table 13.** Linear regression models A1–D1.

| | Model A1 * | | Model B1 * | | Model C1 * | | Model D1 * | |
|---|---|---|---|---|---|---|---|---|
| | Environmental Self-Identity | | Energy-provider Environmental Responsibility | | Personal Norms | | Social Norms | |
| R$^2$ | 0.083 | | 0.025 | | 0.005 | | 0.016 | |
| | Std. Beta | Sig. | Std. Beta | Sig. | Std. Beta | Sig. | Std. Beta | Sig. |
| (Constant) | | 0.5115 | | 0.3047 | | 0.0780 | | 0.1556 |
| Lower secondary school | −0.0023 | 0.9698 | −0.0439 | 0.4812 | −0.1015 | 0.1036 | −0.0815 | 0.1883 |
| Upper secondary school | 0.1081 | 0.0675 | 0.0148 | 0.8104 | −0.1157 | 0.0600 | −0.0819 | 0.1811 |
| BA. MA | −0.0393 | 0.5812 | −0.0275 | 0.7109 | −0.1129 | 0.1282 | −0.1597 | 0.0305 |
| Fulltime employee | 0.0174 | 0.5033 | 0.0080 | 0.7656 | 0.0067 | 0.8056 | −0.0588 | 0.0298 |
| Part-time employee | −0.0199 | 0.3603 | 0.0134 | 0.5489 | −0.0028 | 0.9013 | −0.0531 | 0.0187 |
| Freelancer entrepreneur | 0.0259 | 0.1689 | 0.0084 | 0.6663 | 0.0125 | 0.5246 | −0.0534 | 0.0064 |
| Other employment | 0.0386 | 0.0433 | 0.0315 | 0.1100 | 0.0194 | 0.3304 | 0.0000 | 0.9994 |
| Gender | −0.0683 | 0.0000 | −0.0412 | 0.0122 | −0.0666 | 0.0001 | −0.0588 | 0.0004 |
| Age | 0.2083 | 0.0000 | 0.1374 | 0.0000 | 0.0301 | 0.1737 | 0.0456 | 0.0376 |
| House-type | −0.0634 | 0.0001 | −0.0359 | 0.0276 | 0.0064 | 0.6993 | 0.0042 | 0.7969 |
| Home size (m$^2$) | −0.0682 | 0.0000 | −0.0238 | 0.1315 | 0.0059 | 0.7116 | 0.0141 | 0.3730 |
| Number of appliances | −0.1105 | 0.0000 | −0.0863 | 0.0000 | −0.0277 | 0.0928 | 0.0148 | 0.3642 |

Source: own compilation; (* Sig = 0.000).

The explanation power of Model C1, on analysing the factors influencing personal norms is 0.5%, extremely low. Although gender and the number of electronic appliances seems to be significant factors, the model is too weak.

The explanation power of Model D1, on analysing the factors influencing social norms is 1.6%. Although there are two significant individual factors, the explanation power is too low.

As the next step, we added country variables to the models and present the results of the analysis in Table 14.

According to the models containing the country variables the explanation power of Model A2, on analysing the factors influencing environmental self-identity is 17.5%. The same individual characteristics have a significant effect as in the previous model (A1): gender and age have a significant effect: according to our data males tend to keep themselves more environmentally conscious, and by age environmental consciousness is growing. Looking at home characteristics, we see that home size has no significant effect, while the number of electronic appliances and house-type still has: people living in smaller flats with less electronic appliances tend to keep themselves less environmental-friendly persons. The country of the respondent also has a clear significant effect.

Expanding the model (Model B1) on the factors influencing energy-providers' environmental responsibility by country variables (Model B2) the explanation power increased a little bit: 3.9%. Among the individual characteristics, gender and age have a significant effect, while from the home characteristics, the type of the household, and the number of electronic appliances affect significantly this attitude. According to the analysis, males assume higher the environmental consciousness of the energy providers, than females. This attitude increases by age. Home size and the number of electric appliances significantly affects this attitude. People living in smaller flats and having less

electronic appliances tend to see energy providers less environmentally friendly. Respondents' country also has a significant effect.

**Table 14.** Linear regression models A2–D21.

| | Model A2 * | | Model B2 * | | Model C2 * | | Model D2 * | |
|---|---|---|---|---|---|---|---|---|
| | Environmental Self-Identity | | Energy-Provider Environmental Responsibility | | Personal Norms | | Social Norms | |
| R$^2$ | 0.175 | | 0.039 | | 0.148 | | 0.023 | |
| | Std. Beta | Sig. | Std. Beta | Sig. | Std. Beta | Sig. | Std. Beta | Sig. |
| (Constant) | | 0.9204 | | 0.3920 | | 0.0011 | | 0.1462 |
| Lower secondary school | −0.0234 | 0.6855 | −0.0292 | 0.6440 | −0.0105 | 0.8585 | −0.1229 | 0.0509 |
| Upper secondary school | −0.0472 | 0.4130 | −0.0271 | 0.6669 | −0.0299 | 0.6104 | −0.1336 | 0.0336 |
| BA. MA | −0.0497 | 0.4718 | −0.0145 | 0.8477 | −0.0671 | 0.3388 | −0.1948 | 0.0095 |
| Fulltime employee | −0.0149 | 0.5512 | −0.0067 | 0.8045 | 0.0057 | 0.8219 | −0.0660 | 0.0158 |
| Part-time employee | 0.0142 | 0.4981 | 0.0297 | 0.1886 | 0.0082 | 0.7013 | −0.0552 | 0.0157 |
| Freelancer entrepreneur | 0.0064 | 0.7217 | 0.0003 | 0.9864 | 0.0118 | 0.5207 | −0.0575 | 0.0036 |
| Other employment | 0.0329 | 0.0715 | 0.0264 | 0.1809 | 0.0133 | 0.4759 | −0.0010 | 0.9594 |
| Gender | −0.0698 | 0.0000 | −0.0410 | 0.0126 | −0.0667 | 0.0000 | −0.0603 | 0.0003 |
| Age | 0.1563 | 0.0000 | 0.1200 | 0.0000 | 0.0556 | 0.0075 | 0.0315 | 0.1559 |
| House-type | −0.0584 | 0.0001 | −0.0365 | 0.0254 | −0.0142 | 0.3550 | 0.0081 | 0.6231 |
| Home size (m$^2$) | −0.0261 | 0.0776 | −0.0093 | 0.5590 | −0.0080 | 0.5940 | 0.0227 | 0.1583 |
| Number of appliances | −0.0793 | 0.0000 | −0.0758 | 0.0000 | −0.0364 | 0.0185 | 0.0176 | 0.2876 |
| Italy | 0.3720 | 0.0000 | 0.1150 | 0.0000 | −0.2181 | 0.0000 | 0.0817 | 0.0000 |
| Switzerland | 0.0650 | 0.0000 | −0.0442 | 0.0083 | −0.4164 | 0.0000 | 0.0816 | 0.0000 |

Source: own compilation; (* Sig = 0.000).

The explanation power of Model C1 was low, including country variables increased explanation power to 14.8% in Model C2, analysing the factors influencing personal norms. Among the individual characteristics gender and age have a significant effect: elderly people and males seem to feel that energy saving is moral. the number of electronic appliances also seems to be a significant factor, the more appliances one have the less one agrees with the statements that energy use is a moral issue. Respondents' country also has a significant effect.

The explanation power of Model D2, on analysing the factors influencing social norms is still fairly low: 2.3%. Several individual and household characteristics, and also the respondents' country have a significant effect on it, but the low explanation power leaves few places for analysis. The next question in the analysis is what kind of environmental attitudes groups exist in the four countries.

### 5.3. Attitudes Groups towards Environmental Protection by Countries

Table 15 shows a quaternary typology. The principal components are variables with a zero expected value, with unit standard deviation, that is, measures close to zero mean that the members of the given group in the variable reach the sample mean. Since the average of the original variables is typically between 4.7 and 5.2 (measured on a seven-point scale, where 7 represents the most complete agreement), the sample mean, 0, indicates a certain degree of agreement, so negative numbers do not necessarily mean a clear rejection only that the particular feeling (attitude) is less characteristic of the group than the sample as a whole. A positive number means that a particular emotion is very characteristic of group members.

**Table 15.** Attitude groups towards environmental protection.

| Principal Component | They Are Green for Conscience Reasons | Conscious Greens | Fully Greens | Not Interested in Environmental Protection |
|---|---|---|---|---|
| Environmental self-identity | −0.50 | 0.60 | 0.85 | −1.21 |
| Energy-provider environmental responsibility | −0.22 | 0.40 | 0.85 | −1.21 |
| Personal norms | 0.24 | −0.73 | 0.85 | −1.21 |
| Social norms | −0.11 | −0.16 | 0.85 | −1.21 |
| Number of cases | 1490.00 | 1004.00 | 1637.00 | 1028.00 |
| % | 28.9 | 19.5 | 31.7 | 19.9 |

Source: own compilation.

About the members in the first column, we know that they do not feel environmentally sensitive and that they consider energy service providers sensitive to environmental issues only to a very low degree. They also do not perceive that environmental issues are important to those around them, but they think they should do it for the sake of morality, we call them conscientious environmentalists, 28.98% of the sample may be classified here.

The group in the second column sees no moral reason for environmental protection, feels less receptive to the wider social environment, and at the same time sees itself as the person responsible for environmental protection, and believes that environmental protection is also important to energy service providers. We named this group Conscious Greens, 19.5% of the sample.

For the members of the group in the third column, all four dimensions of environmental protection are very important. They consider themselves to be environmentalists, believe in the environmental responsibility of energy service providers, see no other opportunity from a moral point of view, and believe that protecting the environment is important to those around them. This is the most populous group, fully greens are 31.7% of respondents.

The group in the fourth column is the opposite of the previous one, they have no explicit interest in environmental issues, either morally or socially. We named this group as not interested in environmental protection, 19.9% of the sample belongs to this group.

One of the central questions in this paper is whether there is a significant difference in the attitudes of respondents to energy use in each country. The following table (Table 16) shows that there is a significant correlation between the country of the respondent and attitudes towards energy use and environmental protection.

**Table 16.** The relationship between environmental attitudes and the respondent country on the basis of the examination of std. residuals.

| | Conscious Greens | They Are Green for Conscience Reasons | Not Interested in Environmental Protection | Fully Greens |
|---|---|---|---|---|
| Switzerland | 7.5 | −2.5 | 3.9 | −6.4 |
| The Netherlands | −10.9 | 9.8 | −0.3 | −0.1 |
| Italy | 9.7 | −11.8 | −3.5 | 5.5 |
| Hungary | −7.2 | 5.2 | 1.1 | 0.2 |
| Total | 1036 | 1372 | 753 | 1625 |
| | 21.6% | 28.7% | 15.7% | 34.0% |

Sig = 0.000; Source: Own compilation ($N$ = 4786).

Examination of standard residuals also provides an opportunity to understand which country, the occurrence of which groups is more frequent than expected in the event of independence.

We see that in Switzerland Conscious Greens, who do not see moral reasons for environmental protection, are less likely to be receptive to the broader social environment, but there are more individuals who consider themselves and energy service providers environmentally conscious, while

the proportion of Fully Greens is lower than expected in the event of independence. Swiss respondents are characterized by extremes, as there is a particularly high proportion of those who do not have an interest in environmental protection.

When interpreting the data in Italy, it is worth bearing in mind that the research was conducted in Northern Italy. We can see that the proportion of conscious greens and fully greens is high, and the proportion of greens for conscientious reasons is lower than expected in the event of independence.

Netherlands the proportion of greens for conscience reasons is higher expected in the event of independence, which means that this is a personal, moral issue for people living here, not choosing more environmentally friendly and energy-efficient solutions as a response to social pressure, or because they think that environmental protection is an important issue for them. In contrast, the proportion of conscious greens is expressly low in the Netherlands.

The Hungarian data are basically similar to the Dutch data, the proportion of conscious greens is lower, and the proportion of greens for conscious reasons is higher than expected in the case of independence. Examining the distribution of the groups, we find that the Netherlands and Hungary are somewhat similar.

Our results suggest that while many details remain to be clarified and the need for new explanatory variables is obvious, differences in environmental attitudes across countries may be a key dimension of energy efficiency components in Europe, and more complex analysis should be the focus of future research.

## 6. Discussion and Conclusions

In this paper, we aimed to explore the internal barriers of conscious renewable energy use, energy efficiency and environmental consciousness, by analysing the socio-demographic factors influencing energy use and environmental attitudes in four different European regions. As stated by Cattaneo's theoretical analysis [2], our comparative study confirmed that analyzing consumer attitudes, values, and norms is an important dimension of the economic valuation of renewable energies, and without this, effective incentives for increased use of renewables can be difficult.

We have distinguished four attitudes towards energy use in line with previous findings in the literature [21,22,30,31]: environmental self-identity, energy-provider environmental responsibility, personal norms, social norms. These were created by principal component analysis and then, using the four principal components, we developed a quaternary typology: a separate group of conscious greens, conscientious greens, not interested in environmental protection and fully greens. To separate the effects of the different factors, we conducted a linear regression analysis of the four principal components.

Already, the analysis of the principal components has shown that individuals' attitudes towards (renewable) energy use and environmental awareness differ significantly across countries and across social and demographic variables. In our study, complementing and expanding the results of previous researches [11,12,15,25,28], we have found, that differences in environmental attitudes between countries, which have received less attention in previous analyzes than other factors, may also be important dimensions for understanding and putting into practice European energy efficiency.

Our paper suggests that personal norms, perceived social expectations, and individual environmental awareness are not necessarily interrelated, i.e. internal barriers influencing energy use are diverse. This diversity can be better understood in the light of the social and demographic background of the individual, as demonstrated in the analysis of the principal components. We have also shown that attitudes towards energy use differ between the countries participating in the research and that, accordingly, the distribution of attitudes within the countries studied is different. It now seems necessary to look more closely at what additional socio-demographic indicators can help to better understand the use of energy in individual European countries and regions. Differences between countries are likely to be evident both within countries and in large European regions, and further large-scale quantitative research is needed to analyze this, which may provide a better

basis for European and national energy policies. Further research on the motivating effects of good examples [20] and reference groups [3] is likely to produce significant results and should also include analyzes of social status and habits. The relationship between education, knowledge and the degree to which energy carriers are supplied, and attitudes related to energy efficiency have been confirmed by our analysis, as has been the case in other studies [2]. Further studies are necessary to analyse the connection between social status and attitudes toward energy use.

Our analysis includes data from three developed European countries, and within the less developed Hungary, the more disadvantaged region of the Northern Great Plain. This is a remarkable novelty because it provides an opportunity to measure the relationship between the degree of territorial development and energy efficiency. According to our findings, there is no direct link between the region's economic development and internal barriers related to energy use and energy efficiency. Hungarian respondents are more dissatisfied and distrustful of institutions and service providers than other respondents. Hungarians have an opinion about environmental consciousness that is close to that of the Dutch, while the other internal borders show more Dutch-Swiss and Hungarian-Italian similarities. Further understanding and socio-demographic involvement will be needed in the future to further understand the differences between internal barriers across countries. According to the analysis of the attitude groups, the Hungarian and the Dutch answers are basically similar, and further research may help to understand the deeper reasons. It also seems worthwhile to analyse in the future the policies of each country, as our analysis clearly shows that the population's attitudes towards energy use differ from country to country. Perhaps it would be worth expanding successful environmental psychology researches [3,16,41,42,44] into the analysis of the more complex social status.

In sum, these studies are all the more important because, even when appropriate policies are developed, the question of how attitudes may be translated into real action, how internal barriers affect attitudes over time, and how external barriers shape differences in internal barriers, is a major issue.

**Author Contributions:** Conceptualisation: I.C., I.K., G.B.M., Data curation: I.C., I.K., G.B.M., Formal analysis: I.C., I.K., G.B.M., Investigation: I.C., I.K., G.B.M., Project administration: I.C., G.B.M., Resources: I.C., I.K., Software: G.B.M., Supervision: I.K., G.B.M., Visualisation: I.C., G.B.M., Writing—original draft: I.C., I.K., G.B.M., Writing & editing: I.C., I.K., G.B.M. All authors have read and agreed to the published version of the manuscript.

**Funding:** The study was financed by European Union's Horizon 2020 research and innovation programme under grant agreement No 723791—project PENNY „Psychological, social and financial barriers to energy efficiency". The research was financed by the Higher Education Institutional Excellence Programme of the Ministry of Innovation and Technology in Hungary, within the framework of the Energy thematic programme of the University of Debrecen, and EFOP 3.6.3-VEKOP-16-2017-00007 – "Young researchers for talent" - Supporting career in research activities in higher education"

**Acknowledgments:** We would like to say thank you for our colleague at the PENNY project, Karolina Balogh at the Centre for Social Sciences for her comments, and the three anonymous reviewers.

**Conflicts of Interest:** The authors declare no conflict of interest.

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
