# Peer review of "Environmental Citizenship and Energy Efficiency in Four European Countries (Italy, The Netherlands, Switzerland and Hungary)"

_sustainability, doi:10.3390/su12031154_

Round 1

Reviewer 1 Report

The objectives of the manuscript are clear. They present the topic properly and clearly. The manuscript addresses the current theme. The methods chosen and the conclusions drawn are appropriate. Uniform appearance, well structured.With regard to the content of the manuscript, it is recommended that in the introduction and in the literature the specific features of the countries involved in the study are discussed: the economic, social and cultural environment. Recommendations:

    The methodology needs to be slightly refined. Who did the PENNY 2006 survey? What justified the repetition ten years later. The literature and conclusions warrant further reflection of similar research (such as attitudinal studies) in the countries under study. The text layout of Chapter 4 needs to be improved. Text between tables is justified. It is not advisable to close the chapter with a table. The conclusions are partially justified in comparing the 2006 results with the current ones. The conclusions are partly justified in emphasizing the specific features of each country and explaining it. For example: 5.1. and 5.2. results (dissatisfaction and mistrust of service providers in Hungary, responses reflecting Swiss criticism). The conclusions would in part warrant a brief comparison of the results with those of some other similar studies.

Reviewer 2 Report

The topic is interesting and it is adapt to this journal. The collaboration among several faculties is useful and I think that there is a great work behind the presentation of this work. However, while the presentation is nice in shape, there are few comments and/or suggestions to improve the manuscript. --Clarify better the innovation of this work in the abstract and in the main text. Please add to these sections more general information’s. --The following structure would be preferable based on the Sustainability Microsoft Word template file: 1. Introduction (1.1, 1.2, 1.3.), 2. Materials and Methods (2.1, 2.2., 2.3.), 3. Results and Discussion (3.1, 3.2, 3.3), 4. Conclusions. These sections mixed in the text. https://www.mdpi.com/journal/sustainability/instructions --Please add units ([%], [pc.]) to tables 1 and 2. -Table 4: --Please use the bold function only by country for easier viewing. --Please add units (e.g. [pc.]) to the “Respondent's country” part in table 4. --It is sufficient to use the % unit only in the "Respondent's country" part. For other columns do not have to use the % unit. E.g, table 2: https://www.mdpi.com/2071-1050/11/22/6293/htm --The countries should be better separated for easier viewing. -Tables 5-6: --Table 4 answers. -Table 16: --Use the bold function to the "Total" part. (as in table 15) -Tables 1-6, 9-14, 16: --Please cite the tables in the text before the tables appear. (as in tables 7-8) --Where is the conclusion chapter? Please add to the manuscript a conclusion chapter.

Reviewer 3 Report

Dear Authors,     

Thank you for your submission that discusses the energy related issues and covers four countries.    

The authors’ names and affiliations seems to be omitted from the manuscript. I assume it will be included in later versions.

Section #1: Introduction: The introduction is well-written, and concise as it should be (there is separate literature background section later) and provides good overview of what to expect from rest of the article. The selection of four countries from somewhat different scenarios –itself is an interesting topic.

Line #22: It mentions about the ‘attitude’ towards energy use. What exactly the authors are trying to convey here? The later part of the article goes into details, but at the beginning, the readers may be interested to know what contents lie ahead.

Line #60: The sentences before this presents a very interesting context. Few examples (“such as: …”) of overt and covert barriers may be presented in this context in this paragraph.

Line #89: The first use of the word ‘PENNY’ (except the abstract) should have a reference. The readers are just given a name ‘PENNY Project’. I did Google search and it displayed many results that were for science project using metal penny. Then, I added the country name and was directed to the projects website which had a little more details: “PENNY (Psychological, social and financial barriers to energy efficiency)”. The authors should just put a reference here and direct it to the http://www.penny-project.eu/     

Section #3: The samples (the people participating in this study) is very well described and their various criteria are well documented. In table 6, some horizontal lines may be used to separate the countries. Looking at a number in the table and trying to find the corresponding row and column will be easier.

Line #170: Authors should use reference in this: “Boogen et al. [40] and own compilation”

Section #4: The data was presented nicely in concise format and conveys the responses of the survey quickly for the reader.

Section #5: The task of data interpretation is done well.

Section#6: Title revision suggested. The contents are important and it is treated properly.

Section#7: Concluding discussion wraps up the content properly.

Once again, this reviewer cordially thanks the authors for undertaking this project, processing the survey responses and dedicating their time and energy for writing this article.       

The English language used in this manuscript are of high quality. The sentences are well constructed and the arguments are well articulated
.

I would recommend to publish this paper after the mentioned points have been address.

Sincerely,
The Reviewer
